# The use of mobile apps and fitness trackers to promote healthy behaviors during COVID-19: A cross-sectional survey

**Huong Ly Tong**[1]*, **Carol Maher**[2], **Kate Parker**[3], **Tien Dung Pham**[4], **Ana Luisa Neves**[5,6], **Benjamin Riordan**[7], **Clara K. Chow**[1,8], **Liliana Laranjo**[1,9‡], **Juan C. Quiroz**[10,11‡]

1 Westmead Applied Research Centre, Faculty of Medicine and Health, University of Sydney, Sydney, Australia, 2 Alliance for Research in Exercise, Nutrition and Activity, UniSA Allied Health and Human Performance, University of South Australia, Adelaide, Australia, 3 Deakin University, Geelong, Australia, Institute for Physical Activity and Nutrition (IPAN), School of Exercise and Nutrition Sciences, 4 Royal Melbourne Hospital, School of Computing and Information Systems, The University of Melbourne, Melbourne, Australia, 5 NIHR Imperial Patient Safety Translational Research Centre, Imperial College of London, London, United Kingdom, 6 Centre for Health Technology and Services Research, Department of Community Medicine, Information and Decision in Health, Faculty of Medicine, University of Porto, Porto, Portugal, 7 Centre for Alcohol Policy Research, La Trobe University, Melbourne, Australia, 8 Department of Cardiology, Westmead Hospital, Sydney, Australia, 9 Western Sydney Primary Health Network, Sydney, Australia, 10 Centre for Big Data Research in Health, University of New South Wales, Sydney, Australia, 11 Centre for Health Informatics, Australian Institute of Health Innovation, Macquarie University, Sydney, Australia

‡ These authors are joint senior authors on this work.
* hton5658@uni.sydney.edu.au

## Abstract

### Objectives

To examine i) the use of mobile apps and fitness trackers in adults during the COVID-19 pandemic to support health behaviors; ii) the use of COVID-19 apps; iii) associations between using mobile apps and fitness trackers, and health behaviors; iv) differences in usage amongst population subgroups.

### Methods

An online cross-sectional survey was conducted during June–September 2020. The survey was developed and reviewed independently by co-authors to establish face validity. Associations between using mobile apps and fitness trackers and health behaviors were examined using multivariate logistic regression models. Subgroup analyses were conducted using Chi-square and Fisher's exact tests. Three open-ended questions were included to elicit participants' views; thematic analysis was conducted.

### Results

Participants included 552 adults (76.7% women; mean age: 38±13.6 years); 59.9% used mobile apps for health, 38.2% used fitness trackers, and 46.3% used COVID-19 apps. Users of mobile apps or fitness trackers had almost two times the odds of meeting aerobic physical activity guidelines compared to non-users (odds ratio = 1.91, 95% confidence

**Data Availability Statement:** The data that support the findings of this study are openly available at

https://osf.io/wa5p8/?view_only=06a70c1321114dfc8f45bd4e1affca4b.

**Funding:** HLT was supported by the International Macquarie University Research Excellence Scholarship (iMQRES) (Macquarie University funded Scholarship – No. 2018148) and the Australian Government Research Training Program Scholarship. CM is supported by a Medical Research Future Fund Investigator Grant (APP1193862). The funders had no role in study design, data collection and analysis, decision to publish, or preparation of the manuscript.

**Competing interests:** The authors have declared that no competing interests exist.

interval 1.07 to 3.46, $P$ = .03). More women used health apps than men (64.0% vs 46.8%, $P$ = .004). Compared to people aged 18–44 (46.1%), more people aged 60+ (74.5%) and more people aged 45–60 (57.6%) used a COVID-19 related app ($P$ < .001). Qualitative data suggest people viewed technologies (especially social media) as a 'double-edged sword': helping with maintaining a sense of normalcy and staying active and socially connected, but also having a negative emotional effect stemming from seeing COVID-related news. People also found that mobile apps did not adapt quickly enough to the circumstances caused by COVID-19.

## Conclusions

Use of mobile apps and fitness trackers during the pandemic was associated with higher levels of physical activity, in a sample of educated and likely health-conscious individuals. Future research is needed to understand whether the association between using mobile devices and physical activity is maintained in the long-term.

### Author summary

Technologies such as mobile apps or fitness trackers may play a key role in supporting healthy behaviors and deliver public health interventions during the COVID-19 pandemic. We conducted an international survey that asked people about their health behaviors, and their use of technologies before and during the pandemic. Sixty percent reported using a mobile app for health purposes; 38% used a fitness tracker. People who used mobile apps and fitness trackers during the pandemic were more active than people who did not. Women were more likely to use health apps than men, and people aged 45+ were more likely to use COVID-19 apps than people under 45. Differences in app usage based on sex and age indicate that tailored technologies are needed to support different groups. Participants revealed that they had to adapt their use of mobile apps to fit their needs during the highly restricted circumstances caused by COVID-19. Altogether, our findings provide new insights into how mobile apps and devices can deliver health support remotely during a pandemic, and highlight the need for these technologies to adapt to support people's changing needs.

## Introduction

Coronavirus disease 2019 (COVID-19) and subsequent public health measures have drastically impacted lifestyles worldwide and have had adverse effects on health behaviors [1–6]. Several cross-sectional surveys of adults in Australia, the US and UK have reported negative changes in health behaviors and mental health during the pandemic, including reduced physical activity [3,4], unhealthy eating habits and lower diet quality [3,4], increased alcohol consumption [1], and higher prevalence of anxiety and depression symptoms [1,2,6]. In addition to self-reported changes, studies using objective smartphone-based data also showed a decline in daily step count worldwide [5,7]. During the pandemic, the World Health Organization highlighted the importance of maintaining healthy behaviors in the fight against COVID-19 [8]. With restrictions on face-to-face clinical consultations and the strain on health care

systems in delivering patient care, mobile devices were increasingly harnessed to remotely deliver health care support [9,10].

Mobile devices such as mobile apps and fitness trackers [11] can be leveraged to deliver behavior change interventions and might play a role in supporting healthy behaviors during the pandemic. Specifically, mobile apps and fitness trackers can incorporate behavior change techniques (i.e., the active component of an intervention designed to regulate behavior change [12]) that are known to be effective in changing behaviors. Systematic reviews have found that behavior change techniques such as goal setting and self-monitoring of behavior are effective at improving physical activity and diet outcomes [13,14]. Mobile apps or fitness trackers can deliver these behavior change techniques, such as by enabling users to set their own goals, or to self-monitor some behaviors, as demonstrated in prior reviews [15,16]. During the pandemic, mobile apps and fitness trackers can offer unique benefits, by allowing people to access health support remotely and engage in virtual activities (e.g., livestreamed exercise class), in replacement of disrupted in-person activities. Evidence from systematic reviews suggests that under pre-pandemic or 'normal' conditions, mobile apps and fitness trackers can improve physical activity [17–21], diet [17,22], sleep [23], reduce smoking and alcohol intake [22,24,25], and help manage mental health [17,26]. However, little is known about the use of these technologies for health behaviors during the COVID-19 pandemic, and the association between using mobile apps and fitness trackers, and healthy behaviors.

A few studies have examined the use of digital technologies for physical activity and mental health during the pandemic. Specifically, a study of Google Trends showed an increase in searches for physical activity and exercise in Australia, the US and the UK [27]. An analysis of App store data in the US showed an increase in downloads of mental health apps [28]. Cross-sectional surveys found that the use of digital platforms (e.g., streaming services, mobile apps) was associated with higher physical activity levels [29–31]. While this evidence is promising, the scope was limited to physical activity and mental health and did not explore other behaviors (e.g., diet, smoking, alcohol intake) that are important to maintain good health during the pandemic. Moreover, existing research has not examined the use of fitness trackers, which have been known to have a positive impact on health behaviors [18,20,21]. Thus, there remain gaps in understanding how a range of mobile devices were being used for physical and mental wellbeing during the pandemic, and the association between usage and health behaviors.

In addition to supporting healthy behaviors, mobile devices have also been leveraged to deliver public health interventions during the pandemic. Specifically, mobile apps have been developed for COVID-19 purposes, such as to support contact tracing [9], self-management of symptoms, or home monitoring [32–34]. Despite rapid growth in the number of COVID-19 mobile apps, little is known about their adoption, with preliminary evidence suggesting that specific subgroups (e.g., older people) are more likely to adopt such apps [35]. It is important to better understand how different subgroups might adopt COVID-19 apps, to inform public health strategies and policy makers in their response to the pandemic.

To address these gaps, we conducted a cross-sectional survey to examine use of mobile apps and fitness trackers to support health behaviors (i.e., self-reported physical activity, diet, sleep, smoking, alcohol consumption), mental wellbeing, and public health interventions (e.g., COVID-19 apps) during the pandemic.

The secondary aims of the study were to examine:

1. Whether using mobile apps and/or fitness trackers was associated with healthy behaviors,

2. What was the adoption of COVID-19 related apps (i.e., mobile apps designed specifically for COVID-19), and

3. Whether specific subgroups showed a higher use of COVID-19 related apps and mobile apps and fitness trackers for health-related purposes.

## Methods

### Study design

This study is a cross-sectional survey that examined the use of mobile apps and fitness trackers for health behaviors and public health interventions during the COVID-19 pandemic. The reporting adheres to the STrengthening the Reporting of OBservational studies in Epidemiology (STROBE) guideline for cross-sectional studies [36] (S1 Appendix). Ethical approval was granted by Macquarie University's Human Research Ethics Committee (Approval number: 52020674017063). All participants provided electronic written consent prior to participation (S2 Appendix).

### Settings and participants

An anonymous online survey was hosted on the Qualtrics platform [37]. The study was advertised via various channels, including social media (Facebook, Twitter, LinkedIn, Instagram, Reddit), public posters (e.g., at parks, libraries, university campus), and research institute networks (e.g., email lists, university website). In our social media advertisements, we also asked people to share the study with their networks (e.g., re-tweet on Twitter), in order to expand the geographical scope of the study. Study recruitment was self-selected, i.e., interested individuals could click on the survey link, upon which they were provided with the study information and provided an electronic written consent prior to participation. Eligible study participants were adults aged over 18 years who were proficient in English. We followed published heuristics for sampling for behavioral research and aimed to recruit at least 500 participants into the study [38]. The survey was open from start of June to end of September 2020 to achieve the targeted sample size.

### Context

During the data collection period (June–September 2020), the World Health Organization assessed the global risk of COVID-19 to be very high [39]. The number of infected cases globally increased from over 10 million [40] to 32.7 million [41] during this period, with vastly different infection rates amongst countries. Public health policies across countries varied considerably with respect to lifestyle restrictions such as lockdown measures, travel restrictions, and mask mandates [42,43]. It is worth noting that during June–September 2020, a few countries had started to ease lifestyle restrictions (e.g., Australia, UK, Canada) [43].

### Survey development and measures

Existing COVID-19 surveys [44–46] were reviewed to inform the wording and structure of the present survey. Subsequently, a draft survey was prepared and reviewed independently in three rounds to establish face validity. Specifically, in round one, a draft survey was prepared by the first author and reviewed by a clinician and a computer science expert, with revisions made accordingly. In round two, the survey was sent out to three experts in digital health and behavioral research for feedback, and revised accordingly. Finally, the revision made in round two was reviewed again by a clinician prior to being finalized. A copy of the Qualtrics survey can be found in S2 Appendix.

## Measures

**Demographic characteristics.** Participants reported their age (years), gender (female, male, other, prefer not to say), highest level of education completed (primary school, high school, vocational training, bachelor's degree, postgraduate degree), country of residence, and whether they had medical conditions that required regular medical care or medication (yes, no).

**Health behaviors.** Health behaviors including physical activity, diet, smoking and alcohol consumption during the pandemic were self-reported. Participants were asked how many minutes of moderate-to-vigorous physical activity they completed each week. Participants were considered to have adhered to the recommended levels of aerobic physical activity if they self-reported at least 150 minutes of moderate-to-vigorous physical activity in a week, based on the World Health Organization's guidelines [47].

Participants self-reported daily servings of vegetables and fruits. Participants were considered to have adhered the recommended intake of vegetables and fruits if they self-reported consuming at least five servings of vegetables and fruits in a day, based on the World Health Organization's recommendation [48]. Participants also reported the number of standard drinks they typically have in a week, their smoking status (yes, no) and number of cigarettes smoked in a day. Examples of moderate-to-vigorous physical activity, fruit and vegetable servings, and standard alcoholic drink servings were provided.

**The use of mobile apps and fitness trackers for health behaviors.** The survey contained 20 questions about participants' usage of mobile apps (including health apps, general apps, and social media apps) and fitness trackers to support health-related purposes before and during the COVID-19 pandemic. In the survey, health-related purposes were defined as staying active, eating healthily, sleeping better, reducing/stopping smoking and alcohol drinking, and managing mental wellbeing, and it was specified that the focus was not on chronic disease management (e.g., monitor blood glucose, medication reminders). Usage status during the pandemic was classified into three groups: current users, past users and never-users, based on existing literature [30,31,49]. The definition of usage status is provided in Box 1. Additionally, participants were asked to indicate the extent to which they agreed with the usefulness of technologies in supporting different health behaviors. These items were measured using a five-point Likert scale, ranging from strongly disagree to strongly agree. The survey also contained three optional, open-ended questions to collect qualitative data on how participants used mobile apps, fitness trackers, and other technologies to support health behaviors and mental wellbeing during the COVID-19 pandemic.

## COVID-19 related apps

The survey included two questions about whether people used COVID-19 related apps (i.e., mobile apps created specifically for use during the COVID-19 pandemic), and for what purposes (e.g., for contact tracing, symptom checking).

## Data analysis

Quantitative data were analyzed using R version 4.0.4 [50–52]. Descriptive statistics, including frequencies and percentages, were generated for categorical variables; means and standard deviations (SD) were generated for continuous variables. Two logistic regression models were used to examine the association between 1) the use of mobile apps and fitness trackers and adherence to aerobic physical activity guidelines, and 2) the use of mobile apps and adherence to fruit and vegetable consumption guidelines. Specifically, one logistic regression model included adherence to aerobic physical activity guidelines as the outcome variable, and the

Box 1: Classification based on technology usage during the pandemic*

| Usage status | Definition |
|---|---|
| Current users | People who were currently using mobile apps or fitness trackers for health purposes during the pandemic |
| Past users | People who used mobile apps or fitness trackers for health purposes in the past, but were not currently using them during the pandemic |
| Never-users | People who never used mobile apps or fitness trackers for health purposes |

**Note:** Classification based on [30,31,49]. [30,31] classified participants into users and non-users. We modified this classification to include current users, and broke non-users into past and never-users based on [49] to provide more granularity in usage patterns.

independent variables were current use of mobile apps or fitness trackers, whether participants used an app or tracker before COVID-19 (as a proxy for interest in technology before COVID-19), and whether participants started using a new app or tracker since COVID-19. Another model included adherence to fruit and vegetable consumption guidelines as the outcome variable, and the independent variables were current use of mobile apps, whether participants used a mobile app before COVID-19, and whether participants started using a new app since COVID-19. Both models were adjusted for factors selected a priori, including age, gender, education, and the existence of current medical conditions. Odds ratios (OR) and 95% confidence intervals (CI) were reported. Post-hoc sensitivity analyses were conducted to include only Australia-based participants, given the large proportion of this group in the sample.

Subgroup analyses were conducted to explore to explore whether age and gender subgroups were more likely to use mobile apps for health-related purposes or COVID-19 related apps. These subgroups were chosen based on the literature, as previous cross-sectional surveys have found that app usage might differ by age and gender [30,35]. Specifically, Thomas et al found that COVID-19 app downloads appeared to increase with age, with the 65+ age group having the highest proportion of downloads [35]. Additionally, Parker et al also found that more women than men used digital platforms for their physical activity during the pandemic [30]. Chi-square tests were used for categorical data. When the assumption of chi-square test was violated, Fisher's exact test was used instead. The significance level for all statistical tests was set at $P < .05$, two-tailed.

Qualitative data (from free-text responses) were analyzed using thematic analysis [53] in NVivo 12 [54] to explore the different ways people used technologies to maintain health and wellbeing during the pandemic. Integration of results was conducted after quantitative and qualitative analyses were completed, through embedding of the data. Integration is presented throughout the Discussion section.

## Results

### Sample description

While 554 people consented to participation, two were under 18, and thus, were not eligible. In total, 552 participants (mean age 38±13.6 years, 76.6% women) were included in data analysis. Responses were recorded from 32 countries, with most participants (382/549, 69.6%) living in Australia. The majority (359/552, 65%) had completed a

**Table 1. Sample characteristics of survey respondents.**

| Characteristics | N[a] | Percentage (%)[b] or Mean (SD) |
|---|---|---|
| **Age (years)**; range: 18–80 | | 37.8 (13.6) |
| 18–29 | 180 | 32.6% |
| 30–39 | 172 | 31.2% |
| 40–49 | 89 | 16.1% |
| 50+ | 111 | 20.1% |
| **Gender** | | |
| Female | 423 | 76.6% |
| Male | 120 | 21.7% |
| Other | 5 | 0.9% |
| Prefer not to say | 4 | 0.7% |
| **Education level** | | |
| Primary School | 2 | 0.4% |
| High School | 41 | 7.4% |
| Vocational training | 24 | 4.3% |
| Undergraduate bachelor's degree | 126 | 22.8% |
| Postgraduate degree | 359 | 65% |
| **Country of residence** | | |
| Australia | 382 | 69.6% |
| USA | 52 | 9.5% |
| Vietnam | 35 | 6.4% |
| UK | 23 | 4.2% |
| Canada | 7 | 1.3% |
| Others[d] | 50 | 9.1% |
| **Current medical conditions** | | |
| Yes | 156 | 28.8% |
| No | 385 | 71.2% |
| **Moderate-to-vigorous physical activity**[d] (minutes/week); range: 0–840 | 510 | 164 (152) |
| **Vegetable consumption**[d] (number of daily servings); range: 0–25 | 511 | 2.71 (1.94) |
| **Fruit consumption**[d] (number of daily servings); range: 0–16 | 511 | 1.71 (1.53) |
| **Smoker**[d] | | |
| Yes | 16 | 3% |
| No | 525 | 97% |
| **Cigarettes per week**[d]; range 1–20 | 16 | 0.21 (1.47) |
| **Alcohol consumption**[d, e] (number of standard drinks/week); range 0–60 | 510 | 3.06 (5.33) |

Notes

[a]Total number in each row might not add up to 554 due to missing responses

[b]Sums may not equate to 100% due to rounding

[c]S3 Appendix includes a detailed breakdown of country of residence

[d]Self-reported data

[e]One extreme value 6450 was excluded.

postgraduate degree, and 71.1% (385/541) reported having no current medical condition requiring regular care or medication. The self-reported average weekly time spent in moderate-to-vigorous physical activity was 164 (SD 152) minutes. The average vegetable and

**Table 2. The use of mobile apps for health-related purposes before and during COVID-19.**

| Mobile app usage | n/N (%) |
|---|---|
| **App usage status during the pandemic** | |
| Current users | 302/504 (59.9%) |
| Past users[a] | 103/504 (20.4%) |
| Never-users[b] | 99/504 (19.6%) |
| **Self-reported changes in app usage during COVID-19** | |
| Used app more | 192/401 (47.8%) |
| Used app the same amount as before COVID-19 | 176/401 (43.9%) |
| Used app less | 33/401 (8.2%) |
| **Top 5 most popular apps used for health purposes** | |
| Pre-COVID | YouTube (85/263, 32.3%) |
|  | Facebook (82/263, 31.2%) |
|  | Apple Health (80/263, 30.4%) |
|  | Instagram (69/263, 26.2%) |
|  | Fitbit (67/263, 25.5%) |
| During COVID[c] | Zoom (54/163, 33.1%) |
|  | YouTube (37/163, 22.7%) |
|  | Facebook (19/163, 11.7%) |
|  | Calm (18/163, 11%) |
|  | Fitbit & Houseparty (10/163, 6.1%) |
| **Purposes for app usage during COVID-19** | |
| To stay active | 248/298 (83%) |
| To eat healthily | 77/298 (26%) |
| To sleep better | 78/298 (26%) |
| To reduce/quit smoking | 2/298 (1%) |
| To reduce/quit alcohol | 3/298 (1%) |
| To connect with other people | 109/298 (37%) |
| To manage mental health | 99/298 (33%) |
| **Physical activity purposes for app usage during COVID-19** | |
| To track activity levels | 196/246 (79.7%) |
| To join a live class | 74/246 (30.1%) |
| To follow an exercise video | 148/246 (60.1%) |
| For social aspects (to compete or share progress) | 83/246 (33.7%) |

[a]People who used apps consistently (e.g., use an app more than 5 times) for health purposes in the past, but were not currently using them during the pandemic

[b]People who never used app for health purposes

[c]New apps adopted for health purposes during COVID-19

fruit consumption reported by participants were 2.7 and 1.7 daily servings, respectively. Most of the sample (525/541, 97%) were non-smokers. The average alcohol consumption was reported as 3 drinks per week. The sociodemographic and health characteristics of the study sample are presented in Table 1.

## Technology use for health behaviors and mental wellbeing during COVID-19

**Mobile apps.** Regarding participants' app usage habits, 59.9% (302/504) were currently using apps for health purposes during the pandemic (i.e., current users) (Table 2). Amongst the current app users, 77.8% (235/302) consistently used mobile apps for their health before COVID-19. A greater proportion of women were current app users than men (64.0% vs

46.8%, *P* = .004, S4 Appendix provides more details on subgroup analyses). The most popular apps used for health purposes during the pandemic were general and social media apps (e.g., Zoom, Facebook, Youtube), which were not purposely built to promote health behavior change (Table 2).

Compared to pre-pandemic times, nearly half (192/401, 47.8%) used mobile apps more frequently for health purposes during the COVID-19 pandemic (Table 2). Forty percent (164/401, 40.9%) started using a new mobile app for health-related purposes since the outbreak of COVID-19.

During the COVID-19 pandemic, the most reported health purpose of app usage was to stay active (248/298, 83%) (Table 2). Amongst those who used apps for physical activity, the majority used them to track activity levels (196/246, 79.7%), or to follow an exercise video (148/246, 60.1%) (Table 2). Over two-third of participants (203/298, 68.1%) used mobile apps for more than one health purpose during the COVID-19 pandemic. Compared to men, a greater proportion of women used mobile apps to stay active (48% vs 36.7%, *P* = .02) and to connect with other people (22.7% vs 9.2%, *P* = .004, S4 Appendix).

Regarding the perceived usefulness of mobile apps for health, 59.4% (232/390) of participants agreed that mobile apps helped them incorporate more activity in their days; 43.5% (167/384) agreed that mobile apps helped them manage their mental wellbeing. Compared to men, a greater proportion of women found mobile apps helpful for managing their mental wellbeing (80.6% vs 63.2%, *P* = .04, S4 Appendix).

**Fitness trackers.** Over a third of participants (188/492, 38.2%) were current users of fitness trackers, 19.3% (95/492) were past users, and 42.7% (210/492) had never used fitness trackers for their health. The median length of usage for current and past users was 2 years (range 1 month—10 years). Forty-eight percent of responders (237/492, 48.1%) mentioned that they had used fitness trackers before the pandemic. Amongst those who used trackers before the pandemic, the most popular trackers used pre-COVID were Fitbit, and Apple Watch. Since the COVID-19 outbreak, 5.1% of respondents (25/492) started using a new fitness tracker.

During the pandemic, the most common reasons for using fitness trackers were to track different measurements (e.g., distance run or walked, heart rate), and to receive reminders to move. Over half (147/274, 53.6%) agreed that fitness trackers helped them incorporate more activity in their daily lives.

**The association between technology usage and healthy behaviors.** People who currently used a mobile app or fitness tracker during the pandemic had almost two times the odds of meeting aerobic physical activity guidelines (OR = 1.91, 95% CI 1.07 to 3.46) compared to non-users (Table 3). Whether participants used mobile apps or fitness trackers before COVID-19, and whether participants started using a new app or tracker since COVID-19 were also statistically associated with meeting aerobic physical activity guidelines. Specifically, people who started using a new app or tracker since COVID-19 had 1.7 times the odds of meeting aerobic physical activity guidelines than people who did not (OR = 1.66, 95% CI 1.06 to 2.61) (Table 3). People who had used mobile apps or trackers before COVID-19 had more than 2 times the odds of meeting aerobic physical activity guidelines than non-users (OR = 2.32, 95% CI 1.36 to 4.02). Mobile app usage was not associated with meeting fruit and vegetables consumption guidelines (OR = 0.97, 95% CI 0.53 to 1.76) (Table 3).

Given the large proportion of Australia-based participants in our sample, we conducted a sensitivity analysis with this subgroup (S5 Appendix). The sensitivity analysis showed that current app or tracker usage was no longer statistically associated with meeting aerobic physical activity guidelines (OR = 1.63, 95% CI 0.79 to 3.43). Age, whether participants used an app or tracker before COVID-19, and whether participants started using a new app or tracker since

**Table 3. Adjusted odds ratios (OR) and 95% confidence intervals (CI) for the associations between 1) adherence to aerobic physical activity guideline and use of mobile apps or fitness trackers; 2) adherence to fruit and vegetable consumption guideline and use of mobile apps.**

| Variables | Odds ratio (95% CI) of adherence to aerobic physical activity guideline[a] | p-values | Odds ratio (95% CI) of adherence to fruit and vegetable guideline[b] | p-values |
|---|---|---|---|---|
| **Age** | 1.02 (1, 1.04) | *.02* | 1.01 (0.996, 1.03) | .14 |
| **Gender** | | | | |
| Female | 1 (reference level) | | 1 (reference level) | |
| Male | 1.41 (0.84, 2.40) | .20 | 0.77 (0.45, 1.28) | .32 |
| **Education** | | | | |
| High school | 0.76 (0.31, 1.85) | .55 | 0.52 (0.20, 1.25) | .16 |
| Vocation training | 0.65 (0.20, 1.98) | .44 | 0.90 (0.28, 2.87) | .86 |
| Undergraduate degree | 0.78 (0.47, 1.29) | .33 | 0.43 (0.25, 0.71) | *< .001* |
| Postgraduate degree | 1 (reference level) | | 1 (reference level) | |
| **Current medical condition** | | | | |
| Yes | 0.60 (0.38, 0.94) | *.03* | 0.75 (0.47, 1.17) | .21 |
| No | 1 (reference level) | | 1 (reference level) | |
| **Current app or tracker usage[c]** | | | | |
| Yes | 1.91 (1.07, 3.46) | *.03* | 0.97 (0.53, 1.76)[c] | .91 |
| No | 1 (reference level) | | 1 (reference level) | |
| **Whether an app or tracker was used pre-COVID[c]** | | | | |
| Yes | 2.32 (1.36, 4.02) | *.03* | 1.20 (0.73, 1.97)[c] | .47 |
| No | 1 (reference level) | | 1 (reference level) | |
| **Whether a new app or tracker was used since COVID[c]** | | | | |
| Yes | 1.66 (1.06, 2.61) | *.03* | 1.30 (0.83, 2.04)[c] | .26 |
| No | 1 (reference level) | | 1 (reference level) | |

[a]Participants were considered to have adhered to aerobic physical activity guideline if they self-reported doing at least 150 minutes of moderate to vigorous physical activity in a week

[b]Participants were considered to have adhered to fruit and vegetable consumption guideline if they self-reported having at least 5 servings of fruits and vegetables in a day

[c]The model exploring the link between technologies and fruit and vegetable consumption only considered app usage, not fitness trackers.

COVID-19 were statistically associated with meeting aerobic physical activity guidelines. Mobile app usage was also not associated with meeting fruit and vegetable consumption guidelines in this subgroup (OR = 1.08, 95% CI 0.52 to 2.27).

**COVID-19 related apps.** Less than half of the participants (235/508, 46.3%) used a COVID-19 related app. Of those that used COVID-19 related apps, most used country-specific apps (e.g., COVIDSafe in Australia). The main purpose of using COVID-19 related apps was to support contact tracing. Twelve percent (59/508, 11.6%) used COVID-19 related apps for more than one purpose, most often to support contact tracing and get COVID-19 information.

Use of COVID-19 related apps differed by age and whether they were currently using mobile apps for their health. Compared to people aged 18–44, a larger proportion of people aged 60+ (74.5% versus 46.1%) and a larger proportion of people aged 45–60 (57.6% versus 46.1%) used a COVID-19 related app (*P* < .001, S4 Appendix). Compared to never-users, a greater proportion of current users (50.3% vs 35.3%) and past users (47.6% vs 35.3%) of mobile apps for health used COVID-19 related apps (*P* = .034, S4 Appendix).

**Table 4. Illustrative quotations for qualitative findings.**

**Maintaining a sense of normalcy and social connections**

1. "[Using mobile apps] has provided accountability to myself to do at least a little something of tracked exercise every day. I was furloughed from work, so [I] lost structure to my day and being able to track helped maintain a sense of structure." (F, 28, US)

2. "My use of technology has been extremely helpful in bringing some sense of normalcy into my life. Since I have had limited ability to recreate outside like normal, it has helped me adjust the way that I maintain my fitness by providing me more workout options to do in my apartment (exercise videos, biking on trainer instead of outside, etc.), track progress so I can envision how it would compare to outside, and provide entertainment/ambiance of being outside (scenic videos/nature sounds)." (F, 28, US)

3. "As a family, we have started using the Couch to 10K app to train for a 10K run together. This has helped us stay in touch and keep us motivated." (F, 49, Australia)

4. "Platforms such as WhatsApp, Zoom and Facetime have allowed me to maintain connections with my friends and family. This has been essential for mental health." (F, 42, Australia)

**Technologies as a double-edged sword**

5. "Social media serves as diversion or entertainment and source of information as well during the pandemic." (F, 41, Philippines)

6. "Keeping away from some of the apps made me less stressed. I was not checking any update on COVID19 patients. I avoided all those unsolicited health advices that bombard Facebook/YouTube in form of ads." (F, 39, Australia)

7. "[My fitness tracker] has given me a guilty feeling that my exercise routine is reduced. [I feel] guilty when I see my counts barely reaching my expected goal due to working and not leaving my office." (F, 37, Australia)

**Desired features of technology**

Adaptability

8. "I am someone who has [. . .] solidly incorporated several fitness apps into my daily life. BUT during COVID my use of them almost stopped [. . .]. I was exercising less in solid blocks of a single activity (e.g., at the gym) and cooking more from scratch, both of which are more complicated to log. The impact of apps on my health experience any other time has been a great and positive one but that actually changed during COVID, as they weren't designed for the highly restricted behavior people were forced into." (F, 27, Australia)

9. "During COVID, I used physical activity apps to avoid passing by COVID-19 hotspots/where confirmed cases have been." (M, 43, Vietnam)

10. "My use of apps has reduced as I took responsibility of my own fitness—walking. Because I could only walk in the same area, I developed my own challenges along the route." (F, 61, Australia)

Gamification

11. "I've also used games like Nintendo Switch ring fit and just dance for activity at home." (F, 38, UK)

12. "Now I still use the Qantas Wellbeing app to track my steps (& you can 'compete' with friends in weekly challenges)." (F, 31, Australia)

Notes: The bracket provides gender, age, and country of residence. F: female, M: male.

**Qualitative results.** The most common and central themes from the responses to open-ended questions are described below and comprised: maintaining a sense of normalcy and social connections; technologies as a double-edged sword; desired features of technology. S6 Appendix includes demographic details of the subset of participants who answered each of the open-ended questions.

**Maintaining a sense of normalcy and social connections.** Participants mentioned that during the pandemic, mobile devices has allowed them to maintain a routine despite the disruption caused by COVID-19, and maintain a sense of normalcy, which in turn gave them motivation to exercise (Table 4, quotes 1–2). Additionally, most participants mentioned that technologies helped them stay socially connected with their family and friends, which alleviated some emotional stress and allowed them to share their fitness progress (Table 4, quote 3–4).

**Technologies as a double-edged sword.** Participants cited both positive and negative effects from the use of technologies, especially social media, during the COVID-19 pandemic. On one hand, social media allowed people to stay updated with COVID-19 news (Table 4, quote 5). On the other hand, participants also mentioned that the high volume of COVID-19 news could cause information overload and emotional stress (Table 4, quote 6). Similarly,

when talking about fitness trackers, some participants indicated negative emotions associated with self-monitoring, as their physical activity had declined due to COVID-19 circumstances (Table 4, quote 7).

**Desired features of technology.** There were two subthemes within the area of desired features of technology: adaptability and gamification. Participants mentioned that while technologies had been helpful, one key thing missing was the adaptability of technologies to the unprecedented circumstances caused by COVID-19 (Table 4, quote 8). Consequently, several mentioned that they took the initiative to repurpose existing health apps to serve their needs during COVID-19 pandemic (Table 4, quotes 9–10). Many participants across different ages also valued gamification features of technologies (e.g., competition, exercise challenges, exercise role-playing games), which helped them to incorporate fitness into their life with an element of fun and enjoyment (Table 4, quotes 11–12).

## Discussion

### Principal results

Our study found that 60% of participants used mobile apps and 38% used fitness trackers for health behaviors during June–September 2020. People who used mobile apps or fitness trackers during the pandemic were more likely to self-report meeting recommended levels of aerobic physical activity than non-users. A greater proportion of women used apps for their health during the pandemic than men. Additionally, 46% of respondents self-reported using COVID-19 apps. Specific subgroups such as people aged 45+ and current or past users of mobile apps for health purposes were more likely to use COVID-19 related apps. We note that these subgroup analyses based on age and gender are exploratory in nature and should be confirmed in future research. The generalizability of our quantitative findings is limited, given our sample of highly educated individuals who might have been more health-conscious, and had better access and more inclined to use technologies. Qualitative findings complemented quantitative findings by showing while mobile devices helped maintain a sense of normalcy, there were potential negative effects of using technologies (e.g., stress and information overload from seeing COVID-19 information on social media, guilt when seeing low activity levels), which might have impacted users' motivation and continued use of mobile devices. Our participants highlighted the need for technologies to adapt to changing circumstances.

### Impact of mobile devices on health behaviors

Our results are consistent with existing literature showing that users of mobile apps and other digital technologies seem to be more active than non-users during the pandemic [29–31,55]. Uniquely, by adjusting our model to variables related to 'previous use of mobile devices before COVID' and 'adoption of new apps or trackers during the outbreak', we found these were associated with adherence to physical activity guidelines. It is possible that the physical activity benefits observed in our study are influenced by an overrepresentation in our sample of health-conscious and tech-adopting people. Future research is needed to understand how mobile devices can extend its reach and benefit other groups beyond the typical highly motivated and 'worried-well' adopters [56]. A sensitivity analysis including only Australia-based participants found that current mobile app or tracker usage was not associated with adherence to physical activity guidelines. It is possible that the smaller sample size made it difficult to detect the difference. Given the inconsistency between the primary and sensitivity analyses, the potential physical activity benefits associated with mobile devices observed in our findings should be interpreted with caution, and future research is needed to ascertain the potential impact of mobile devices on health behaviors.

Our qualitative data highlight the need for mobile apps and fitness trackers to adapt quickly to the changing circumstances of human lives, especially in health crises like COVID-19. Given the disruption to normal routines and closure of exercise and health facilities, people might need additional, or different types of support to maintain healthy behaviors, which is difficult to accommodate by mobile apps and devices based on static algorithms. With recent development in artificial intelligence and machine learning, mobile apps and devices can collect information about its users (including users' behaviors, context or preferences) to continuously *adapt* their content, timing and delivery, and personalize their support to suit the person's needs [57,58].

## Differences in app usage between genders

Findings suggested that a greater proportion of women used mobile apps during the pandemic than men. Specifically, women were more likely to use apps to support physical activity and to connect with others, and more likely to report apps as useful for mental health. It is worth noting that this gender difference is based on a subgroup analysis and is exploratory in nature. However, we also note that our finding is in line with previous research reporting higher use of digital platforms for physical activity amongst women [30]. There are several possible explanations for this observed gender difference. Research has shown that during the pandemic, women reported increased overeating [4] and less physical activity than men [59], and heightened stress from taking on more caring or home-schooling responsibilities [1,59–62]. Thus, women might have needed additional support and turned to mobile devices to support their wellbeing. Another possible explanation is linked to the type of health activities that can be accommodated in health apps. Research has suggested that women were more likely to engage in directed activities (e.g., exercise classes [63,64]), which could be delivered online more easily, compared to competitive sports usually done by men [63]. Future research is needed to explore how the adoption of mobile devices might differ by gender and how to design health interventions to reduce the existing gender differences in adoption.

## Adoption and usage of COVID-19 related apps

Only 46.3% of our participants used a COVID-19 related app. Previous research has reported uptake ranging from 20% [65,66] to 40% [35,67] amongst European countries and Australia. Given that the most common purpose is contact tracing, this low uptake is concerning as digital tracing apps rely on a high adoption rate to work effectively [9]. Research has suggested that the reasons for low uptake are mainly privacy and functionality concerns (e.g., battery drain, apps not working as intended) [35]. This indicates the need to improve the functionality of digital tracing apps, as well as public health communication regarding the privacy protections of tracking technologies [68]. Our study found a greater proportion of people aged 60+, and people aged 45–60 used COVID-19 related apps compared to those less than 45 years. This is in line with previous research which suggests that the higher uptake in older adults might be related to concerns about their vulnerability to COVID-19 [35]. This trend highlights the need for public health communication to also target younger populations to ensure a high adoption rate in this subgroup. It is worth noting that since 2021, some countries (e.g., Australia) have made 'signing-into' venues mandatory, usually through a 'check-in' function in government apps to support contact tracing. Thus, since the completion of this study, it is likely that the use of these government apps for COVID-19 purposes have increased. Furthermore, given the exploratory nature of this subgroup analysis, future research is needed to confirm potential age differences in COVID-19 app uptake.

## Strengths and limitations

A strength of our study is the mixed-methods design, including qualitative, open-ended questions, which allowed us to acquire a deeper exploration of users' perspectives. However, the results must be interpreted considering some limitations. While face validity was established through multiple co-authors independently reviewing the survey draft, the survey questions were not formally assessed for criterion or content validity, and the survey was not pilot tested. Health behaviors were assessed through self-report. We assessed the impact of technologies on only aerobic physical activity and the intake of fruits and vegetables. To enable a more comprehensive analysis on the link between technologies and physical activity and diet, future research should collect data on other types of activity (e.g., muscle strengthening exercises) and food groups (e.g., salt or sugar intake). We were not able to examine the link between technologies usage and alcohol intake and smoking because only a small percentage of our sample used technologies for these purposes. While our sampling was worldwide, the majority of participants resided in Australia. As a large proportion of participants were women, and had high level of education, this might bias our findings and affect the generalizability to other population groups. Previous surveys have reported a similarly high participation rate from women and people with higher education levels [1,3,4,30]. The survey was conducted online and proficiency in English was required, which might have precluded participation from non-English speaking individuals and those lacking access to the Internet. Finally, our findings are also impacted by common limitations of survey research—self-reported answers and self-selection sampling method. This might have led to sampling bias, social desirability bias, or recall bias, which affect the generalizability of the findings and the reliability of the responses.

## Implications

Mobile apps and fitness trackers seem promising in promoting physical activity during the COVID-19 outbreak. Potential improvements on these technologies from users' perspectives should focus on personalization and adaptability, such as allowing for higher customization of content delivered and a better ability to support people's changing needs. This is in line with previous research which suggests that personalization can increase user engagement with mobile devices [69]. By leveraging recent advances in big data and artificial intelligence [58], mobile devices may be able to provide more in-time, personalized support to users. Future research is needed to investigate whether the engagement with health apps and devices is sustained post-COVID, and robust clinical trials are needed to ascertain their objective benefits for preventative health, including physical activity and other health behaviors.

Our findings may be influenced by the large proportion of highly educated individuals who might be more health-conscious and have access to technologies more easily than other population groups. Previous research has described this phenomenon as the "digital divide" [70,71], which can widen existing social inequalities. The benefits of mobile apps and devices would be limited if they can only reach high socioeconomic status groups. Thus, efforts must be made to bridge this gap in technology adoption, such as through increasing access, promoting collaborative and inclusive design, and improving digital literacy [70,71].

## Conclusion

Our study found a positive impact of mobile apps and fitness trackers on physical activity during the pandemic, in a sample of likely health-conscious and technology-inclined individuals. Qualitative data revealed the lack of flexibility of mobile apps and devices and highlighted the

need for these technologies to adapt quickly to changes in life circumstances. Future research should assess the use of mobile apps and fitness trackers post-COVID, and whether these technologies provide objective benefits to health behaviors.

## Supporting information

**S1 Appendix. STROBE checklist.**
(DOC)

**S2 Appendix. Survey.**
(PDF)

**S3 Appendix. Country of residence breakdown by the number of responses and %.**
(DOCX)

**S4 Appendix. Subgroup analyses.**
(DOCX)

**S5 Appendix. Sensitivity analyses in the Australia sub-sample.**
(DOCX)

**S6 Appendix. Demographic information of participants who responded to open-ended questions.**
(DOCX)

## Author Contributions

**Conceptualization:** Huong Ly Tong, Liliana Laranjo, Juan C. Quiroz.

**Data curation:** Huong Ly Tong, Tien Dung Pham.

**Formal analysis:** Huong Ly Tong, Tien Dung Pham.

**Investigation:** Huong Ly Tong, Carol Maher, Ana Luisa Neves, Benjamin Riordan, Liliana Laranjo, Juan C. Quiroz.

**Methodology:** Huong Ly Tong, Carol Maher, Ana Luisa Neves, Benjamin Riordan, Liliana Laranjo, Juan C. Quiroz.

**Project administration:** Huong Ly Tong.

**Resources:** Huong Ly Tong.

**Software:** Huong Ly Tong, Tien Dung Pham.

**Supervision:** Huong Ly Tong, Liliana Laranjo, Juan C. Quiroz.

**Validation:** Huong Ly Tong.

**Writing – original draft:** Huong Ly Tong.

**Writing – review & editing:** Huong Ly Tong, Carol Maher, Kate Parker, Tien Dung Pham, Ana Luisa Neves, Benjamin Riordan, Clara K. Chow, Liliana Laranjo, Juan C. Quiroz.

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
