## [Decision Letter · Decision Letter 0]

27 Apr 2022

PDIG-D-21-00140

The use of mobile apps and fitness trackers to promote healthy behaviors during COVID-19: A cross-sectional survey

PLOS Digital Health

Dear Dr. Tong,

Thank you for submitting your manuscript to PLOS Digital Health. After careful consideration, we feel that it has merit but does not fully meet PLOS Digital Health's publication criteria as it currently stands. Therefore, we invite you to submit a revised version of the manuscript that addresses the points raised during the review process.

Please address the comments of the reviewers attached below. Several reviewers raised concerns regarding the testing and validation of the survey, lack of generalisability due to self-selection biases and the international nature of the survey, especially in the context of differences in Covid-19-related measures and health behaviour recommendations.

Please submit your revised manuscript by . If you will need more time than this to complete your revisions, please reply to this message or contact the journal office at digitalhealth@plos.org. Please include the following items when submitting your revised manuscript:

We look forward to receiving your revised manuscript.

Kind regards,

Laura M. König

Academic Editor

PLOS Digital Health

Journal Requirements:

1. Please update your Competing Interests statement. If you have no competing interests to declare, please state: “The authors have declared that no competing interests exist.”

Additional Editor Comments (if provided):

Reviewers' comments:

Reviewer's Responses to Questions

**Comments to the Author**

1. Does this manuscript meet PLOS Digital Health’s publication criteria? Is the manuscript technically sound, and do the data support the conclusions? The manuscript must describe methodologically and ethically rigorous research with conclusions that are appropriately drawn based on the data presented.

Reviewer #1: Yes

Reviewer #2: Yes

2. Has the statistical analysis been performed appropriately and rigorously?

Reviewer #1: Yes

Reviewer #2: Yes

Reviewer #3: Yes

3. Have the authors made all data underlying the findings in their manuscript fully available (please refer to the Data Availability Statement at the start of the manuscript PDF file)?

Reviewer #1: Yes

Reviewer #2: Yes

Reviewer #3: Yes

4. Is the manuscript presented in an intelligible fashion and written in standard English?

Reviewer #1: Yes

Reviewer #2: Yes

Reviewer #3: Yes

5. Review Comments to the Author

Reviewer #1: This is an interesting research. 

I have highlighted a few comments to improve the manuscript.

1. Was the survey items/questions validated?

2. Line 146: What is the rationale for the study duration?

3. Why was snowball sampling explicitly used?

4. Line 153: specify the period (date) rather than "during this time."

5. Line 189: 5-point Likert scale?

6. Line 249: What is the frequency of Mobile App usage (e.g., once a week, daily, monthly)?

7. Suggest putting survey items/questions in a table for easy understanding.

8. Define the current, past, and never users categories (include references). Some studies use light or heavy users.

9. Did you identify the gamification features that users were interested in?

10. Line 362-363: Suggest to detail the negative effects.

11. Important to highlight the negative and positive effects on user motivation and adherence?

12. Can gamification positively affect the adoption and post-adoption of the COVID-19 apps?

13. Interested in knowing if these apps were theory-based and their relevance to the present of future study.

14. The results can be expanded to compare with similar studies. An interesting study to consider is:

User Engagement and Abandonment of mHealth: A Cross-Sectional Survey (2022). In Healthcare (Vol. 10, No. 2, p. 221). MDPI.

Thank you.

Reviewer #2: - The self-selection bias in this sample is so significant that I have to question the added value for research. But this is not limited to sociodemographic patterns. How valid are statements about subgroups of different usage behavior in a sample that seem to be strongly overrepresented by participants with higher health and fitness orientation? 

- Another substantial issue is the inclusion of participants from such diverse national backgrounds. Covid 19 incidences and distancing rules varied massively around the globe within the corresponing survey period. Amongst other reasons, this was obvious, at least because of the seasonal variation in pandemic events in the northern vs. southern hemisphere and completely different policy measures (e.g., USA vs. Australia). Therefore, I would strongly recommend limiting the analyses to the Australian subpopulation.

- Furthermore, I would recommend not to focus the analyses of so-called “subgroups” only on differential frequencies of app usage depending on sociodemographic characteristics. The authors should consider applying methods for exploring subgroups within the sample instead, e.g., cluster analysis or latent class analyses.

- I think using an app tracker before the pandemic is a strongly confounded proxy for interest in technology, as this may also indicate increased health and fitness orientation of the participants in particular. (For example, only 3% reported being smokers , with an average of 0.2 cigarettes a week!).

Reviewer #3: Thank you for this interesting manuscript describing technology usage and healthy behaviors during COVID-19.Please find below my comments:

Major comments:

1) I would like to know if you pilot-tested the survey. If not, could you please describe the methodology that you followed in order to create a validated survey?

2) "Diet was assessed by self-report of daily servings of vegetables and fruits."

Why did you assess only those two food categories? I would appreciate it if you could provide data on the rest of the food groups.

3) In Table 2, how did you define "more or less" in the usage of the app? 

4)"Mobile app usage was not associated with meeting fruit 296 and vegetables consumption guidelines (OR = 0.97, 95% CI 0.53 to 1.76) (Table 3)"

Since you are including data from people coming from different countries and those ones have different guidelines, how did you end up with adherence or no to fruit and vegetable guidelines?

5)Could you maybe group the qualitative data? To my opinion, the quotations do not add a lot to the manuscript.

Minor comments:

Please check Table 1 to ensure that all categories reach 100%

6. PLOS authors have the option to publish the peer review history of their article (what does this mean?). If published, this will include your full peer review and any attached files.

**Do you want your identity to be public for this peer review?** For information about this choice, including consent withdrawal, please see our Privacy Policy.

Reviewer #1: Yes: Abdulsalam Salihu Mustafa

Reviewer #2: No

Reviewer #3: No

**Comments to the Author**

1. Does this manuscript meet PLOS Digital Health’s publication criteria? Is the manuscript technically sound, and do the data support the conclusions? The manuscript must describe methodologically and ethically rigorous research with conclusions that are appropriately drawn based on the data presented.

Reviewer #3: Yes

---

## [Decision Letter · Decision Letter 1]

14 Jul 2022

The use of mobile apps and fitness trackers to promote healthy behaviors during COVID-19: A cross-sectional survey

PDIG-D-21-00140R1

Dear Ms Tong,

We are pleased to inform you that your manuscript 'The use of mobile apps and fitness trackers to promote healthy behaviors during COVID-19: A cross-sectional survey' has been provisionally accepted for publication in PLOS Digital Health.

Best regards,

Laura M. König

Academic Editor

PLOS Digital Health

Reviewer Comments (if any, and for reference):

Reviewer's Responses to Questions

**Comments to the Author**

1. If the authors have adequately addressed your comments raised in a previous round of review and you feel that this manuscript is now acceptable for publication, you may indicate that here to bypass the “Comments to the Author” section, enter your conflict of interest statement in the “Confidential to Editor” section, and submit your "Accept" recommendation.

Reviewer #1: All comments have been addressed

Reviewer #2: All comments have been addressed

2. Does this manuscript meet PLOS Digital Health’s publication criteria? Is the manuscript technically sound, and do the data support the conclusions? The manuscript must describe methodologically and ethically rigorous research with conclusions that are appropriately drawn based on the data presented.

Reviewer #1: Yes

Reviewer #2: Yes

3. Has the statistical analysis been performed appropriately and rigorously?

Reviewer #1: Yes

Reviewer #2: Yes

4. Have the authors made all data underlying the findings in their manuscript fully available (please refer to the Data Availability Statement at the start of the manuscript PDF file)?

Reviewer #1: Yes

Reviewer #2: Yes

5. Is the manuscript presented in an intelligible fashion and written in standard English?

Reviewer #1: Yes

Reviewer #2: Yes

6. Review Comments to the Author

Reviewer #1: I am happy with all the responses from the Authors. The quality of the manuscript has been enhanced.

Reviewer #2: I would like to thank the authors for their efforts related to the revision of the manusciprt and the additional analysis undertaken to address the issues highlighted in my previous review. I have no further comments to add.

7. PLOS authors have the option to publish the peer review history of their article (what does this mean?). If published, this will include your full peer review and any attached files.

**Do you want your identity to be public for this peer review?** For information about this choice, including consent withdrawal, please see our Privacy Policy.

Reviewer #1: **Yes: **Abdulsalam Salihu Mustafa

Reviewer #2: No
